# *VvMYBPA2* Regulated the Accumulation of Flavan-3-ols though Forming a Trimeric Complex in 'Zaoheibao' Grape

**Changmei Liang** [1,†], **Jianyong Guo** [2,†], **Mingxiang Chen** [2], **Xuehui Zhang** [2], **Guorong Zhang** [2], **Pengfei Zhang** [2], **Jinjun Liang** [2] and **Pengfei Wen** [2,*]

[1]   College of Information Science and Engineering, Shanxi Agricultural University, Taigu, Jinzhong 030801, China
[2]   College of Horticulture, Shanxi Agricultural University, Taigu, Jinzhong 030801, China
[*]   Correspondence: wenpengfei@126.com; Tel.: +86-136-3354-8791
[†]   These authors contributed equally to this work.

**Abstract:** Flavan-3-ols are monomers of Proanthocyanidins (PAs), which are important polyphenolic compounds in grapes. Previous studies had shown that *VvMYBPA2* was closely related to grape flavan-3-ol monomers biosynthesis, but its regulatory network is still unclear. Here, we found that the contents of (+)-catechin and (−)-epicatechin, the enzyme activities of anthocyanidin reductase (ANR) and leucoanthocyanidin reductase (LAR) and the expression of *VvANR* and *VvLAR1* were increased in the *VvMYBPA2* overexpression grape leaves compared to the control. It was proved that VvMYBPA2 protein interacted with VvWDR1 and VvWDR1 protein interacted with VvMYC2 by yeast two-hybrid (Y2H) and bimolecular fluorescence complementation (BiFC). The promoters of *VvANR* and *VvLAR1* were bound by VvMYBPA2 using yeast one-hybrid (Y1H) assay. These results suggested that VvMYBPA2 could form a trimeric complex with VvWDR1 and VvMYC2 and jointly regulated the expression of flavan-3-ol monomers related genes *VvANR* and *VvLAR1*, thereby affecting the enzyme activities of ANR and LAR and ultimately regulating the contents of flavan-3-ols.

**Keywords:** grape; flavan-3-ols; proanthocyanidins; *VvMYBPA2*; MBW complex

## 1. Introduction

Proanthocyanidins (PAs), namely condensed tannins (CTs), are formed from flavan-3-ol monomers and their polymers, which could be decomposed to produce anthocyanins when heated in an acidic environment [1]. It plays an important role in grape stress resistance and human health. PAs are commonly found in many different tissues of plants and play important roles in plant defense against pathogens and other diseases, regulation of seed dormancy, pigmentation and stomatal opening. Numerous studies had shown that dietary PAs had the potential to prevent various diseases in the human body [2]. The contents of PAs affected the taste, flavor and organoleptic properties of grapes [3–6]. It is closely related to the shelf life and color stability of wine [7] and is also an important health care component in wine [8]. Therefore, elucidating the molecular mechanism of PA biosynthesis in grapes is of great significance to improve its stress resistance and fruit quality.

PAs take the dihydroquercetin produced by the phenylalanine metabolic pathway as the substrate, under the catalysis of dihydroflavonol reductase (DFR), to generate leucoanthocyanidins. Then catechin is produced under the action of leucoanthocyanidin reductase (LAR). Epicatechin is generated under the action of anthocyanin synthase (ANS) and anthocyanidin reductase (ANR). They form dimers, trimers and multimers with leucoanthocyanidins under the action of condensed enzyme (CON), namely Pas [9,10]. Two key structural genes, *ANR* and *LAR*, unique to the PA synthesis pathway [11,12], and the LAR and ANR enzymes encoded by them catalyze the formation of flavan-3-ol monomers (+)-catechin and (−)-epicatechin, respectively. One ANR gene (*VvANR*) and two LAR genes (*VvLAR1*, *VvLAR2*) were identified in grapes, and related studies confirmed





the specificity of *VvANR*, *VvLAR1* and *VvLAR2* for grape PA synthesis [3,13]. In the previous research in our laboratory, silencing *VvANR* in grape leaves reduced the content of (−)-epicatechin and (−)-epigallocatechin [14].

The most important transcription factors regulating PA biosynthesis are the MYB family genes. For example, *AtTT2* [15], *MdMYB9/11* [16], *GsMYB60* [17], *DkMYB19/20* [18], *PuMYB134* [19], *GmTT2A/B* [20], etc. Previous studies had shown that the MYB protein was usually involved in the formation of the MYB-bHLH-WD40 (MBW) complex, which regulated flavonoid biosynthesis in turn [21]. AtTT2 (MYB) interacted with the transcription factors AtTT8 (bHLH) and AtTTG1 (WD40) to form the MBW complex, which induced the accumulation of PAs in *Arabidopsis* seed coat by directly regulating the transcription level of PAs synthesis structural gene *AtBANYULS* (*AtBAN*) [22]. MBW complexes also existed in other horticultural plants involved in their PA biosynthesis, such as FaMYB9/FaMYB11-FaTTG1-FabHLH3 [23]; CsMYB60-CsbHLH42/MYC1-CsWD40 [17]; DkMYB2/DkMYB4-DkMYC1-DkWDR1 [24], etc.

*VvMYBPA1* was the first factor found to regulate PA synthesis in grape [25], and ectopic expression of *VvMYBPA1* resulted in increased PA synthesis in *Arabidopsis* and complemented the phenotype of the *Arabidopsis* mutant *tt2*, although there was no homologous gene of *VvMYBPA1* in *Arabidopsis*. While *VvMYBPA2* was mainly expressed in young fruits and peels [26], its expression pattern was consistent with the accumulation of PA during grapevine development. *VvMYBPA2* was highly homologous to *AtTT2*, and heterologous overexpression also increases PA biosynthesis of *Arabidopsis*. Compared with *VvMYBPA1*, *VvMYBPA2* had a preferential role in the synthesis and accumulation of PAs in berry pericarp [26]. VvWDR1 (WD40 protein) [27] and VvMYC2 (bHLH protein) [27,28], which had high similarity to AtTTG1 [22] and ZmMYC2 [29], respectively, had been proved to be involved in the biosynthesis of PAs or anthocyanidins in grapes. MBW or MBWW complexes in grapes had been shown to be involved in their flavonoid biosynthesis, for example, VvMYBA1-VvWDR1-VvMYC1 regulated grape anthocyanin synthesis [30]; the VvMYB5-VvMYC1-WDR1-WRKY26 complex was associated with accumulation of flavonoids in grapes [31,32]. However, there were few studies on the regulatory mechanism of MBW complex for grape flavan-3-ols biosynthesis.

In this study, the positive regulation of *VvMYBPA2* on grape flavan-3-ols biosynthesis was further demonstrated by transient overexpression experiments. The VvMYBPA2 protein had been proved that it interacted with VvWDR1 and VvMYC2 to form a complex by Y2H and BiFC assays, and the promoters of *VvANR* and *VvLAR1* were bound by VvMYBPA2 through Y1H assay. These results confirmed the role of the MBW complex in grape flavan-3-ols biosynthesis and enriched our understanding of the regulatory network of flavan-3-ols biosynthesis.

## 2. Materials and Methods

### 2.1. Test Materials

Leaves were collected from *Vitis vinifera* L. cv. 'Zaoheibao' grown at the horticultural station of Shanxi Agricultural University, with plant spacing and row spacing of 1.0 m and 2.5 m, respectively; vines were planted in hedgerows and managed routinely. New shoots with stems were collected as infection materials at 3 pm on 23 September 2021 year.

Seeds of *Nicotiana benthamiana* were planted in an incubator with a 16 h/8 h photoperiod and 25 °C temperatures.

The vectors pCAMBIA1300, pCAMBIA1300-35S-YFPc and pCAMBIA1300-35S-YFPn were donated by Professor Xu Jin of Shanxi Agricultural University. Yeast strains AH109, Y1Hgold, vector pCAMBIA1300-35S-GFP, pGBKT7, pGADT7 and pAbAi were maintained in our laboratory. The primer sequences used are listed in Appendix A.

### 2.2. Subcellular Localization

The full-length CDS of *VvMYBPA2* without the termination codon was fused to the pCAMBIA-1300-GFP plant expression vector that was driven by the CaMV35S promoter



containing the green fluorescent protein (GFP) reporter gene to produce the fusion vector 35S::*VvMYBPA2*-GFP. The recombinant vector and control vector 35S::GFP were then introduced into tobacco (*Nicotiana benthamiana*) leaves by agroinfiltration. The cells of transformed tobacco leaves were observed via a laser scanning confocal microscope (Leica TCS SP8, Wetzlar, Germany).

### 2.3. Transient Expression of Grape Leaves

The full-length CDS sequence of *VvMYBPA2* was inserted into pCAMBIA1300 overexpression vector to construct 35S::*VvMYBPA2* and transformed into *Agrobacterium tumefaciens* GV3101 competent cells. A total of 120 healthy grape shoots were collected, and 60 shoots were set as a group. The grape leaves were infected by vacuum for 10 min, the infection solution was gently wiped from the leaves and leaves were placed in an LED light incubator with light intensity of 2000 lx, relative humidity of 60–70% and temperature of 25 °C for light and dark alternate culture of 16 h and 8 h.

### 2.4. DMACA Staining

According to the method of Li et al. [33], the infected grape leaves were soaked in ethanol–glacial acetic acid (GAA) solution (3:1, v/v) for discoloration, then rinsed with 75% ethanol for 12 h and washed with distilled water. Finally, the leaves were dyed in 0.6% DMACA solution (6N HCl:methanol = 1:1) for 2 min, observed and photographed. The darker staining indicated the higher content of PAs.

### 2.5. Epicatechin and Catechin Content

According to the method of Liang et al. [14], we accurately weighed grape leaves (0.3 g) into a centrifuge tube, added 3 mL of 70% methanol and extracted by ultrasonic for 25 min. Centrifuged at 12,000 rpm for 10 min, the supernatant was collected and centrifuged repeatedlyand filtered through a 0.45 μm organic microporous membrane. After rotary evaporation, 1 mL of pure water was added for extraction, and 2 mL of ethyl acetate was added for extraction, and the upper layer was taken and extracted with 1 mL pure methanol. Then, the epicatechin and catechin contents were determined by HPLC (main instruments include autosampler, chromatographic column and detector, etc.) (Thermo Fisher, Waltham, MA, USA) after filtration through a 0.22 μm organic microporous (Shengze Technology Co., Ltd., Tianjin, China) membrane.

Mobile phase B was 1.3% glacial acetic acid; the detection wavelength was 280 nm; the injection volume was 5 μL; the flow rate was 1.0 mL·L$^{-1}$; and the column temperature was 30 °C.

### 2.6. Determination of ANR and LAR Enzyme Activity

The determination of ANR enzyme activity was performed according to Liang et al. [14]: we accurately weighed grape leaves (0.4 g) into a centrifuge tube and added 2 mL phosphate-buffered saline (PBS) (pH = 7.4), mixed and let stand for 10 min. Then, the mixture was centrifuged at 1500 rpm at 4 °C for 15 min, and the supernatant was taken as the crude enzyme solution. An accurately weighed total of 1.0 mL of crude enzyme solution, Tris-HCl (pH = 7.7), cyanidin chloride (CYA) and reduced coenzyme II tetrasodium salt (NADPH·Na$_4$) was added, and the mixture was reacted at 45 °C for 20 min. HCl–methanol and vanillic aldehyde–methanol were added. The ANR enzyme activity was mainly determined by measuring the content of epicatechin as follows: U = mg EC·mg$^{-1}$·prot$^{-1}$·min$^{-1}$.

The LAR enzyme activity was determined according to Wen et al. [34] with slight modifications: We put accurately weighed grape leaves (0.4 g) into a centrifuge tube, and 2 mL boric acid buffer solution at pH = 8.8 was added. After mixing via vortex and centrifugation for 15 min at 1500 rpm, the supernatant was taken as the crude enzyme solution. A total of 1.0 mL of crude enzyme solution was accurately weighed, reduced coenzyme II (NADPH), Dihydroquercetin (DHQ) and Tris–HCl (pH = 7.7) were added in turn and then the mixture was reacted at 37 °C for 1 h. Then HCl–methanol and vanillic aldehyde–methanol were added and further reacted at 37 °C for 1 h. The LAR

enzyme activity was mainly determined by measuring the content of catechin as follows: $U = mg\ CAT \cdot mg^{-1} \cdot prot^{-1} \cdot h^{-1}$.

### 2.7. Gene Expression Analysis

Total RNA was extracted by the cetyltriethyammonium bromide (CTAB) method with slight modifications, and the cDNA was obtained by reverse transcription. *VvActin* was used as the internal reference gene. According to the instructions of the $2 \times$ Realtime PCR Supper mix (SYBR green, with anti-Taq), we performed predenaturation at 95 °C for 1 min, followed by 40 cycles of denaturation at 95 °C for 15 s and annealing temperature at 55 °C for 15 s; fluorescence was measured at 72 °C for 30 s, and the relative gene expression was calculated by the $2^{-\Delta\Delta CT}$ method.

### 2.8. Gene Isolation and Sequence Analysis

The ORFs of grape PA biosynthesis regulators (*VvMYBPA2*, *VvWDR1*, *VvMYC2*) were cloned by PCR. Homologous search was performed on the NCBI BLAST server. Molecular evolutionary genetic analysis (MEGA) software version 7 was used to construct phylogenetic tree by adjacency method, and sequence alignment was performed by CLUSTAL X program. Bootstrap analysis is implemented by 500 repetitions. Protein ID: VvMYBPA1, CAJ90831; VvMYBPA2, EU919682; VvMYBPAR, BAP39802; VvMYBA1, BAD18977; VvMYBA2, BAD18978; VvMYB5b, AAX51291; AtTT2, Q9FJA2; MdMYB9, ABB84757; MdMYB11, AAZ20431.1; FaMYB9, AFL02460; ZmC1, 1613412E; DkMYB2, AB503699; PtMYB134, ACR83705; DkMYB4, AB503671; NtMYB2, BAA88222; AtMYB114, AEE34502; AtMYB90, NP_176813; AtMYB75, NP_176057; MrMYB1, ADG21957; AtMYB113, NP_176811; PhAN2, AAF66727; VvMYC1, ACC68685; VvMYCA1, NP_001267954; VvMYC2, NP_001267974; AtTT8, OAO98324; PhAN1, AF260919_1; MtTT8, AKN79606; MdbHLH33, ABB84474; MdbHLH3, ADL36597; ZmMYC2, QDM55339; AtMYC2, NP_174541; PhAN11, AAC18941; AtTTG1, NM122360; NtTTG1, ACJ06978; VvWDR2, ABF66626; VvWDR1, ABF66625; MdTTG1, AAF27919; VcTTG1, MH717246; ZmPAC1, AAM76724; FaTTG1, JQ989287; InWDR1, AB232779; PfWD40, AB059642.

### 2.9. Y2H Assays

The full-length CDS sequences of *VvMYBPA2*, *VvWDR1* and *VvMYC2* were cloned and connected to pGADT7 and pGBKT7 vectors, respectively. The recombinant pGADT7 and pGBKT7 plasmids were co-transformed into yeast AH109 competent cells and then coated on SD/-Trp-Leu defect medium for screening. The positive colonies were inoculated on SD/-Trp-Leu-His-Ade, SD/-Trp-Leu-His-Ade-X-α-gal medium for interaction screening, and the colony growth was observed and recorded. pGADT7-T and pGBKT7-53 were used as positive controls, and pGADT7-T and pGBKT7-Lam were used as negative controls.

### 2.10. BiFC Assays

The CDS sequences of *VvMYBPA2*, *VvWDR1* and *VvMYC2* were cloned and ligated into pCAMBIA1300-35S-YFPc and pCAMBIA1300-35S-YFPn vectors, respectively. The recombinant plasmids were introduced into the competent cells of *A. tumefaciens* GV3101. The cYFP-*WDR1*/nYFP-*MYBPA2*, cYFP-*MYC2*/nYFP-*MYBPA2*, cYFP-*WDR1*/nYFP-*MYC2* and the negative control cYFP/nYFP in the experimental group were injected into the tobacco leaves of the four-week-old tobacco plants, respectively. Two to three days after injection, the laser confocal microscope (Leica) was used to observe and record. DAPI was used for nuclear staining.

### 2.11. Y1H Assays

The cis-acting elements of promoter *VvANR* and *VvLAR1* were predicted according to the PlantCARE online website. The fragments containing MBS elements (100–200 bp) were inserted into a pAbAi vector to construct the pAbAi-*ANR* and pAbAi-*LAR1* vectors. The mutation-related vectors pAbAi-*anr* and pAbAi-*lar1* were constructed by Shang-

hai Biological Engineering Co., Ltd. They were transformed into Y1Hgold competent cells and coated on SD/-Ura medium to screen positive colonies and prepare bait vector competent cells. The pGADT7 empty vector was transformed into pAbAi-*ANR*, pAbAi-*LAR1*, pAbAi-*anr* and pAbAi-*lar1* competent cells, respectively, and coated on SD/-Leu, SD/-Leu/AbA$^{50 \, ng/mL}$, SD/-Leu/AbA$^{75 \, ng/mL}$, SD/-Leu/AbA$^{100 \, ng/mL}$ and SD/-Leu/AbA$^{200 \, ng/mL}$. The concentration of AbA was screened. Finally, the experimental group pGADT7-*MYBPA2*/pAbAi-*ANR*, pGADT7-*MYBPA2*/pAbAi-*LAR1* and negative control pGADT7-*MYBPA2*/pAbAi-*anr*, pGADT7-*MYBPA2*/pAbAi-*lar1* co-transformed yeast Y1Hgold competent state and coated with the corresponding concentration of AbA SD/-Leu screening medium was observed and recorded.

*2.12. Data Processing*

GraphPad Prism 9.0 and TBtools were used for data processing, and SAS 8.0 software was used for significant difference analysis. Photoshop CS6 was used for image processing.

## 3. Results

*3.1. Subcellular Localization of VvMYBPA2*

In order to explore the subcellular localization of grape VvMYBPA2 protein, the subcellular localization vector 35S::*VvMYBPA2*-GFP was constructed and transiently expressed in tobacco leaves, and 35S::GFP was used as a control. The results of laser confocal microscopy showed that the fluorescent green signal in the experimental group was concentrated in the nucleus, while it was dispersed throughout the cells in the empty vector group (Figure 1).

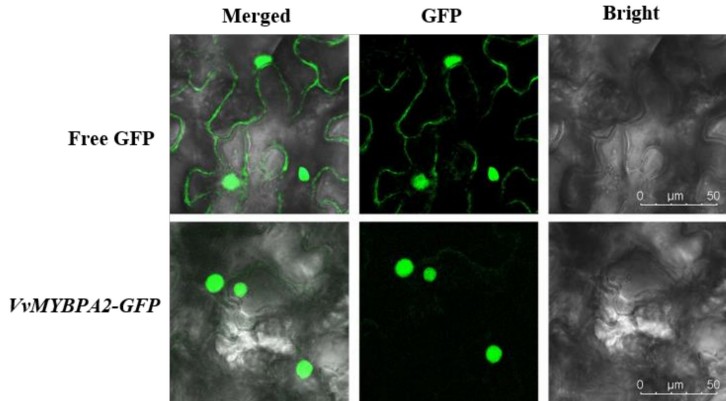

**Figure 1.** Subcellular localization of VvMYBPA2 in tobacco leaf cells. The empty GFP vector was used as a control. The images were taken in the dark field for green fluorescence (**middle**), bright light to demonstrate the morphology of the cells (**right**) and in overlay (**left**).

*3.2. VvMYBPA2 Promoted the Accumulation of PAs in Grape Leaves*

In order to explore whether *VvMYBPA2* could promote the accumulation of PAs in grape leaves, we constructed an overexpression vector 35S::*VvMYBPA2* and used the empty vector pCAMBIA1300 as a control to infect grape leaves by vacuum infiltration. We found significant differences in DMACA staining, which indicated the higher content of PAs, in grape leaves after transient expression (Figure 2a), which indicated that overexpression of *VvMYBPA2* increased the content of PAs in grape leaves. Fluorescence quantitative results showed that the expression of *VvANR* in grape leaves was significantly increased after transient overexpression, the expression of *VvLAR1* was extremely significantly increased and there was no significant difference between *VvLAR2* and the control (Figure 2b). The epicatechin content of the control group was 2.34 μg/g FW, while the content of the overexpression was 5.82 μg/g FW; the catechin content in the control group was 13.27 μg/g FW, while the content after overexpression was 51.06 μg/g FW. It was found that the epicatechin and catechin content were significantly higher (Figure 2c). By measuring the ANR and LAR enzyme activities of the samples, it was found that the ANR enzyme activity

of the grape leaves overexpressing *VvMYBPA2* was significantly increased (Figure 2d), and the LAR enzyme activity was extremely significantly increased (Figure 2e).

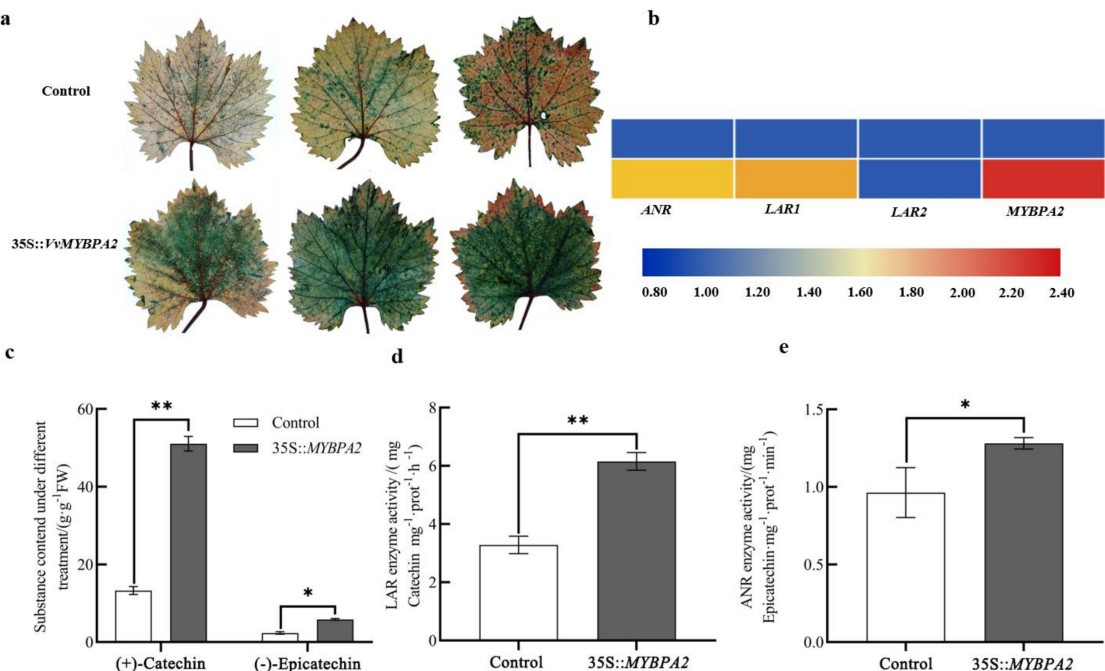

**Figure 2.** Effect of transient overexpression of *VvMYBPA2* in grape leaves. (**a**) DMACA staining results under different treatments. (**b**) Effects of different treatments on gene expression related to proanthocyanidin biosynthesis. (**c**) Effects of different treatments on catechin and epicatechin content. (**d**,**e**) Effects of different treatments on activities of LAR and ANR enzymes. Note: ** indicates extremely significant differences at the level of $p < 0.01$; * indicates significant differences at the level of $p < 0.05$, while ns indicates no significant difference.

### 3.3. Phylogenetic Analysis of Genes Related to Grape PAs Synthesis

Phylogenetic analysis of the deduced amino acid sequences was performed using the neighbor-joining method. Phylogenetic analysis showed that the amino acid sequence of VvMYBPA2, VvMYC2 and VvWDR1 was the most similar to AtTT2, ZmMYC2 and AtMYC2, AtTTG1 and PhAN11, respectively (Figure 3a–c).

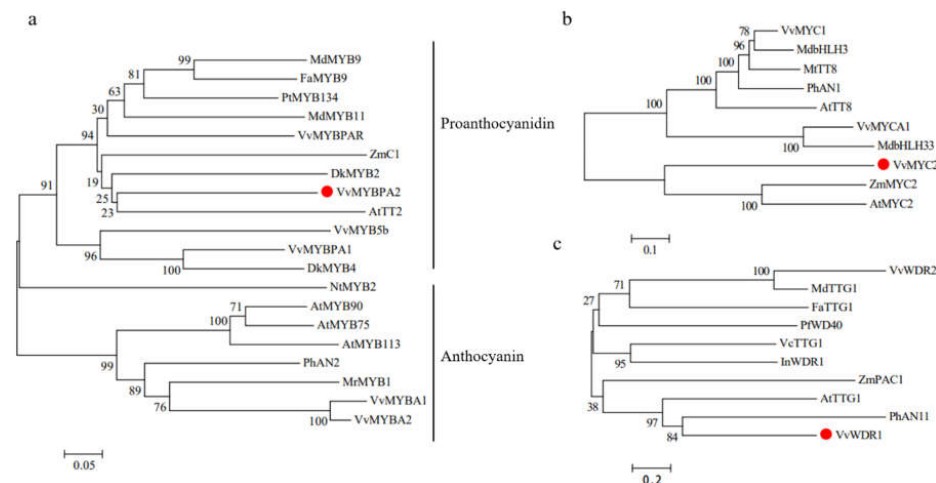

**Figure 3.** Phylogenetic analysis of transcription factors related to grape proanthocyanidin synthesis. (**a**–**c**) means the phylogenetic relationships between MYB, bHLH, WD40 transcription factors with their closest homologues, respectively.

### 3.4. VvMYBPA2 Formed a Trimeric Complex with VvWDR1 and VvMYC2

In order to clarify the interaction between VvMYBPA2 protein, bHLH and WD40, we performed Y2H and BiFC assay. The Y2H results showed that both the positive control and the experimental groups could grow on the four-deficient plate, but only positive control and *VvWDR1/VvMYC2* turned blue on SD/-Trp-Leu-His-Ade/*X-α-Gal* plates (Figure 4a). It indicated that the VvMYBPA2 protein interacted with the VvWDR1 and VvMYC2 proteins at the yeast level. We further verified using a BiFC assay, which showed that in tobacco leaves co-transformed with nYFP-*VvMYBPA2* and cYFP-*VvWDR1*, nYFP-*VvMYC2* and cYFP-*VvWDR1*, yellow fluorescence was detected in the nucleus, but no fluorescence was observed in tobacco leaves co-transformed with nYFP-*VvMYBPA2* and cYFP-*VvMYC2* (Figure 4b). It indicated that VvMYBPA2 could interact with VvWDR1 protein, but not VvMYC2, while VvWDR1 could interact with VvMYC2.

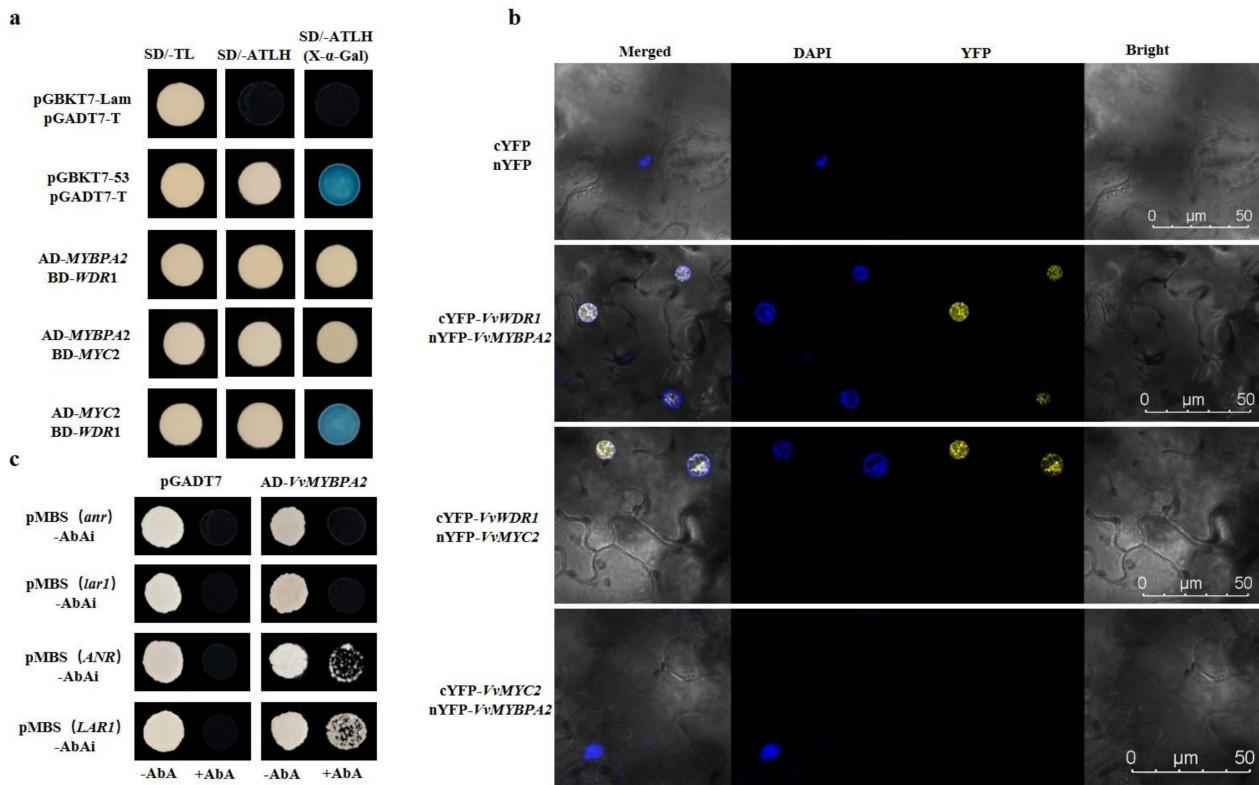

**Figure 4.** Interaction between *VvMYBPA2* and PAs biosynthesis-related genes. (**a**) Y2H assay analyzed the interaction among MYBPA2, WDR1 and MYC2 in vitro. (**b**) The interaction of VvMYBPA2, VvWDR1 and VvMYC2 in tobacco cells was analyzed by BiFC assay. (**c**) The interaction between *VvMYBPA2* and the promoters of *VvANR* and *VvLAR1* was analyzed by Y1H assay.

### 3.5. VvMYBPA2 Directly Activated VvANR and VvLAR1 Promoters

To clarify the binding between *VvMYBPA2* and the promoters of proanthocyanidin biosynthesis-related genes *VvANR* and *VvLAR1*, we carried out Y1H assay. The results showed that both the empty vector and mutant control successfully inhibited the growth on the SD/leu+75 ng AbA plate (Figure 4c), while the experimental group grew on the SD/leu+75 ng AbA plate, indicating that *VvMYBPA2* can directly bind to the MBS acting elements of *VvANR* and *VvLAR1* promoters. Similarly, the expression of *VvANR* and *VvLAR1* genes and related enzyme activities were significantly increased in the transient over-expression test of grape leaves. Therefore, *VvMYBPA2* can directly activate *VvANR* and *VvLAR1* promoters and then affect the accumulation of grape proanthocyanidins.

## 4. Discussion

Research had shown that plant proanthocyanidin biosynthesis was regulated by the TT2 type genes, a large branch of R2R3 MYBs, which belongs to the fifth subgroup [35], including *AtTT2* [15], *MdMYB9/11* [16], etc. Previous localization studies on *VvMYBPA2* homologous proteins AtTT2 [15] and *MdMYB9/11* [16] showed that they were all localized in the nucleus. In this study, the TT2 transcription factor VvMYBPA2 in 'Zaoheibao' grape was cloned, and the subcellular localization-related vector 35S::*VvMYBPA2*-GFP was successfully constructed and injected into tobacco to observe its localization in the nucleus (Figure 1).

Previous studies had found that *VvMYBPA2*, which is highly homologous to *Arabidopsis* TT2, was mainly expressed in the pericarp and leaves of young grape fruit, and its expression pattern was consistent with the accumulation of PAs during grape fruit development [25]. In addition to grapes, poplar [19], alfalfa [36], narcissus [37] and cocoa beans [36] were also found in MYB transcription factors homologous to *Arabidopsis* TT2, and the overexpression of these transcription factors all up-regulated the gene expression levels of LAR and ANR in transgenic plants, eventually leading to the accumulation of PAs. Overexpression of *VvMYBPA2* in grape hairy roots induced its accumulation of PAs and significantly activated *VvANR, VvLAR1* and *VvMYBPA1* [25]. Preliminary research in our laboratory found that the content of PAs in silencing *VvMYBPA2* grape leaves was significantly lower than that of the control, the activities of ANR and LAR enzymes were significantly reduced and the content of flavan-3-ol monomers was reduced. In this study, we found that the contents of (+)-catechin and (−)-epicatechin in grape leaves were significantly increased, ANR and LAR enzyme activities were significantly increased and levels of *VvANR* and *VvLAR1* gene expression were increased after transient overexpression of *VvMYBPA2* (Figure 2).

The R2R3-MYB transcription factor was involved in regulating different branches of flavonoid biosynthesis. To regulate PAs and anthocyanin biosynthesis, MYB transcription factors interacted with basic helix-loop-helix (bHLH) and WDR-repeat (WDR) factors to form ternary complexes to control related structural gene expression [38]. *Arabidopsis* AtTT2-AtTT8-AtTTG1 [22], kiwifruit AcMYBF110-AcbHLH1-AcWDR1 [39], poplar PuMYB134-PubHLH131 [19] and others had been shown to form MBW complexes that were involved in the regulation of anthocyanin and proanthocyanidin biosynthesis. Instead, there were exceptions to the MBW complex model. For example, the Apple WD40 protein MdTTG1 interacted with bHLH but did not interact with the MYB protein to regulate the accumulation of anthocyanins [40]. In grapes, studies had found that MYBA1 interacted with WDR1, and WDR1 interacted with MYC1 to form a complex to regulate the synthesis of grape anthocyanins 'Yan' 73 in tobacco [30]. Studies had shown that WD40 provided a docking platform for MYB-bHLH interaction so that MYB-WD40-bHLH could form a ternary complex to regulate the biosynthesis of anthocyanins [41]. In this study, we found that VvMYBPA2 could interact with VvWDR1 through Y2H and BiFC experiments, but there was no in vivo interaction with VvMYC2. We also observed that VvWDR1 interacted with VvMYC2, which is similar to previous studies. Taken together, VvMYBPA2 and VvMYC2 form a trimeric complex by cooperating with VvWDR1, and affect proanthocyanidin. Previous studies had found that VvWDR1 protein could bind to VvWRKY26 protein [32]. It was speculated that the MBWW complex (VvMYBPA2-VvMYC2-VvWDR1-WRKY26) might be involved in regulating the synthesis of grape proanthocyanidins.

Genes related to proanthocyanidin biosynthesis are divided into two categories: structural genes and transcription factors. Among them, the structural genes directly encode the biosynthetic enzymes in the process of proanthocyanidin synthesis, namely anthocyanin reductase (ANR) and leucoanthocyanidin reductase (LAR), which catalyze the production of epicatechin and catechin monomers, respectively. Generally, transcription factors enhance or inhibit gene expression by binding to specific DNA sequences within the promoter region of the target gene [42]. For example, GsMYB60 [17], DkMYB2/4 [24], MdMYB9/11 [16], PtrBBX23 [43], etc. could activate the expression of proanthocyanidin

biosynthesis-related outcome genes ANR or LAR directly. Grape MYBPAR [44] could also activate PA synthesis-specific genes VvCHS3, VvF3′5′H, VvLAR2 individually. However, some MYB transcription factors must rely on the interaction of bHLH or WD40 proteins to activate the promoters of structural genes, such as *Arabidopsis* AtTT2-AtTT8-AtTTG1 [22] and *strawberry* FaMYB9/FaMYB11-FabHLH3-FaTTG1 [23]. In this study, the Y1H assay showed that VvMYBPA2 could directly bind to the promoters of *VvANR* and *VvLAR1*, thereby regulating the biosynthesis of (+)-catechin and (−)-epicatechin.

## 5. Conclusions

In short, it was found that VvMYBPA2 could form a trimeric complex with VvMYC2 and VvWDR1 in grape and jointly regulated the expression of structural genes *VvANR* and *VvLAR1*, thereby promoting the ANR and LAR enzymatic activities, which ultimately positive regulated (+)-catechin and (−)-epicatechin contents (Figure 5).

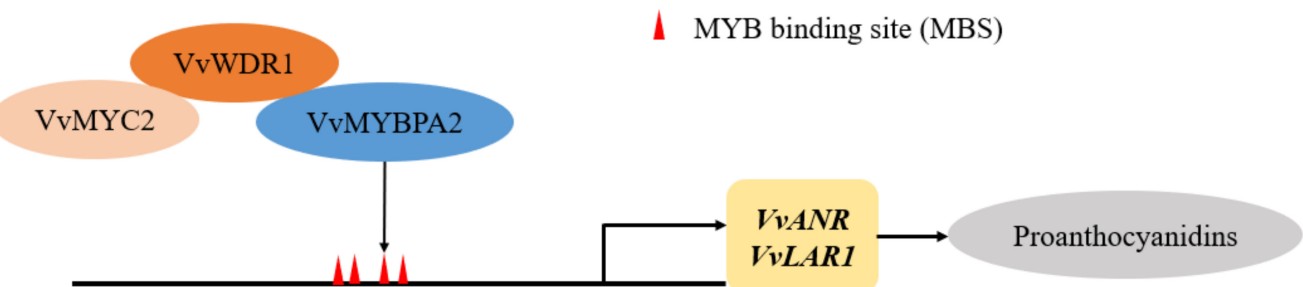

**Figure 5.** Prediction of the biosynthesis pattern of grape PAs regulated by MBW complex.

**Author Contributions:** Conceptualization, J.L. and P.W.; methodology, J.L.; validation, P.Z.; formal analysis, X.Z. and G.Z.; investigation, C.L.; resources, J.G., M.C., J.L. and C.L.; data curation, X.Z.; writing—original draft preparation, J.G.; writing—review and editing, C.L. and J.G.; visualization, C.L.; supervision, P.W.; project administration, J.L.; funding acquisition, J.L. All authors have read and agreed to the published version of the manuscript.

**Funding:** This research was funded by the Biological Breeding Engineering Project of Shanxi Agricultural University (Grant Nos: YZGC113), Youth Scientific Research Project of Shanxi Province (Grant Nos: 20210302124067), Shanxi Province doctoral graduates and postdoctoral researchers come to work in Shanxi to reward scientific research projects (Grant Nos: SXBYKY2021075), and Shanxi Fruit Tree Industry Technology System Post Expert Special Project.

**Institutional Review Board Statement:** Not applicable.

**Informed Consent Statement:** Not applicable.

**Data Availability Statement:** The data presented in this study are available on request from the corresponding author.

**Acknowledgments:** Thanks to Xu Jin for donating tobacco seeds and related vectors.

**Conflicts of Interest:** The authors declare no conflict of interest.

## Appendix A. Information of primers used in PCR

| Primers Name | Primiers Sequence | Accession Number |
|---|---|---|
| GFP-MYBPA2-F | GG<u>GGTACC</u>ATGGGAAGAAGACCTT | EU919682.1 |
| GFP-MYBPA2-R | CG<u>GGATCC</u>TGGGACTTGATTATTTTC | |
| OX-MYBPA2-F | GG<u>GGTACC</u>ATGGGAAGAAGACCTT | EU919682.1 |
| OX-MYBPA2-R | CG<u>GGATCC</u>CTATGGGACTTGATTATTT | |
| RT-*VvActin*-F | CCCCATGCTATCCTTCG | AF369524.1 |
| RT-*VvActin*-R | AGGCAGCTCATAGTTC | |
| RT-*VvANR*-F | AGAACTACAGGAGTTGGGTGAC | DQ129684.1 |
| RT-*VvANR*-R | CCTTGAATTGCTGGCTTG | |
| RT-*VvLAR1*-F | ACGATGTCCGAACACTGAAC | AJ865336 |
| RT-*VvLAR1*-R | TGAACGCCGCTACTACACTC | |
| RT-*VvLAR2*-F | TCTCGACATACATGATGATGTG | AJ8653 |
| RT-*VvLAR2*-R | TGCAGTTTCTTTGATTGAGTTC | |
| RT-*VvMYBPA2*-F | GTGCGATGCTCCAAGGTTAT | EU919682.1 |
| RT-*VvMYBPA2*-R | CAGTGCTGAACTTGAGGGC | |
| BD-*VvWDR1*-F | CG<u>GAATTC</u>ATGGAGAGATCAAGCC | DQ517913.2 |
| BD-*VvWDR1*-R | CG<u>GGATCC</u>CTAAACTTTAAGAAGC | |
| AD-*VvMYC2*-F | GGAATTC<u>CATATG</u>ATGAAAACTGAAATGGG | EF636725.2 |
| AD-*VvMYC2*-R | TCC<u>CCCGGG</u>TTACCCAACTGATGATGAC | |
| BD-*VvMYC2*-F | CG<u>CCCGGG</u>ATGAAAACTGAAATGGGTATG | EF636725.2 |
| BD-*VvMYC2*-R | GGC<u>GTCGAC</u>TTACCCAACTGATGATGAC | |
| BiFC-*MYBPA2*-F | CG<u>GGATCC</u>ATGGGAAGAAGACCTT | EU919682.1 |
| BiFC-*MYBPA2*-R | GC<u>GTCGAC</u>TGGGACTTGATTATTTTCAG | |
| BiFC-*WDR1*-F | C<u>GGATCC</u>ATGGAGAGATCAAGCCT | DQ517913.2 |
| BiFC-*WDR1*-R | GGC<u>GAATTC</u>AACTTTAAGAAGCTGCAGTTT | |
| BiFC-*MYC2*-F | GGC<u>GTCGAC</u>ATGAAAACTGAAATGGGTATG | EF636725.2 |
| BiFC-*MYC2*-R | GG<u>ACTAGT</u>CCCAACTGATGATGACAAAG | |
| Y1H-*ANR*-F | C<u>GAGCTC</u>GTTAGTTGGGAACCATC | DQ129684.1 |
| Y1H-*ANR*-R | GC<u>GTCGAC</u>GCATATCTCAACAGCAG | |
| Y1H-*LAR1*-F | C<u>GAGCTC</u>ACATAAATCCGGCCTAG | AJ865336 |
| Y1H-*LAR1*-R | GC<u>GTCGAC</u>TGACTCACCATTCATGA | |

Note. The underlined part is the enzyme cleavage site.

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
