# Peer review of "VvMYBPA2 Regulated the Accumulation of Flavan-3-ols though Forming a Trimeric Complex in ‘Zaoheibao’ Grape"

_agriculture, doi:10.3390/agriculture12091414_

Round 1

Reviewer 1 Report

In this paper, the molecular mechanism of VvMYBPA2 regulating the accumulation of total flavane-3-ols in 'Zaoheibao' grapes was investigated, and the VvMYBPA2 positive regulation of grape flavane-3-ol monomer content was verified by transient hyperexpression grape leaves. Finally, it was preliminarily proved that VvMYBPA2 can form a trimer with MYC2 and WDR1 to regulate VvANR and VvLAR, and finally regulate the accumulation of flavane-3-ol monomers content through Y2H, BiFC and Y1H. Overall, the experiment was well designed and conducted, the data analysis was proper and results is interesting. however, there are many defects in the text format and pictures, and it needs to be corrected before publication.

1.The 6 pictures in Figure 1 could be more spaced appropriately, and the scale of free GFP is a bit unclear.

2. The font of the absc horizontal coordinate of Figure 2c is different from that of Figures 2d and e, and the semicolon in the figure note is changed to a period.

3. Please explain why DMACA is stained in Figure 2a, and which secondary metabolites are specifically indicated by staining in this paper?

4. The overall writing of the article is relatively standardized, but there are still many minor problems, such as some formatting problems in the references.

Author Response

Dear reviewer

Thank you for your comments concerning our manuscript entitled “VvMYBPA2 regulated the accumulation of flavan-3-ols though forming a trimeric complex in 'Zaoheibao' grape” (agriculture-1895392). Those comments are all valuable and very helpful for revising and improving our paper. Revised portion are marked using “Track Changes” function in the paper. The main corrections in the paper and the responds to the reviewer’s comments are as flowing:

1.The 6 pictures in Figure 1 can be spaced some distances appropriately, and the scale of free GFP is a bit unclear.

A: Figure 1 had been redrawn based on your comments, see Figure 1 for details.

  1. The font of the absc horizontal coordinate of Figure 2c is different from that of Figures d and e, and the semicolon in the figure note is changed to a period.

A: Figure 2 had been redrawn based on your comments, see Figure 2 for details.

  1. Explain why DMACA is stained in Figure 2a, and which secondary metabolites are specifically indicated by staining?

A: DMACA staining is mainly aimed at proanthocyanidins. The darker the color, the higher the proanthocyanidin content. It had been supplemented in the appropriate place in this paper.

  1. The overall writing of the article is relatively standardized, but there are still many minor problems, such as some formatting problems in the references.

A: The references had been re-checked carefully as you suggested, and some formatting issues had been corrected.

Reviewer 2 Report

This work is devoted to the study of role of VvMYBPA2, encoding a R2R3-MYB transcription factor, in regulation of proanthocyanidibs. The authors carried out a large amount of experimental work to prove that VvMYBPA2 could form a trimeric complex with VvMYC2 and VvWDR1 in grape, and jointly regulated the expression of structural genes VvANR and VvLAR1, thereby promoting the ANR and LAR enzymatic activities, which positive regulated (+)-catechin and (-)-epicatechin contents. The manuscript is very well written. The conclusions correspond to the obtained results. It should be noted that the authors used modern methods and approaches in this study.

Decision: - Accept after minor revisions (which the authors can be trusted to make).

- Minor Revisions

1) lines 107, 108, 222, 297, 326: tobacco => tobacco.

2) line 107: agroinfiltration => agroinfiltration

3) line 112: agrobacterium => Agrobacterium tumefaciens

4) line 131: HPLC - which device was used? Company and country of manufacture?

5) line 132: microporous membrane - Company and country of manufacture?

6) line 160: renaturation => annealing temperature

7) line 193: Agrobacterium => A. tumefaciens

8) line 319, 322: kiwifruit, Poplar, Apple => kiwifruit, poplar, apple

9) Figure 5.: the authors should decipher the abbreviations shown in the figure.

Author Response

Dear reviewer

Thank you for your comments concerning our manuscript entitled “VvMYBPA2 regulated the accumulation of flavan-3-ols though forming a trimeric complex in 'Zaoheibao' grape” (agriculture-1895392). Those comments are all valuable and very helpful for revising and improving our paper. Revised portion are marked using “Track Changes” function in the paper. The main corrections in the paper to the reviewer’s comments are as flowing:

1) lines 107, 108, 222, 297, 326tobacco => tobacco.

2) line 107agroinfiltration => agroinfiltration

3) line 112: agrobacterium => Agrobacterium tumefaciens

4) line 131: HPLC - which device was used? Company and country of manufacture?

5) line 132: microporous membrane - Company and country of manufacture?

6) line 160: renaturation => annealing temperature

7) line 193: Agrobacterium => A. tumefaciens

8) line 319, 322kiwifruit, Poplar, Apple => kiwifruit, poplar, apple

A: Detailed revisions and additions had been made to your questions 1-8.

9) Figure 5.: the authors should decipher the abbreviations shown in the figure.

A: We had re-mapped Figure 5 and identified the relevant shapes.
